# The Characteristics of COVID-19 Vaccine-Associated Uveitis: A Summative Systematic Review

**DOI:** 10.3390/vaccines11010069

**Published:** 2022-12-28

**Authors:** Yasmine Yousra Sadok Cherif, Chakib Djeffal, Hashem Abu Serhan, Ahmed Elnahhas, Hebatallah Yousef, Basant E. Katamesh, Basel Abdelazeem, Abdelaziz Abdelaal

**Affiliations:** 1Faculty of Medicine, University of Algiers, Algiers 16311, Algeria; 2Tanta Research Team, El-Gharbia 31516, Egypt; 3Department of Ophthalmology, Hamad Medical Corporations, Doha 576214, Qatar; 4Faculty of Medicine, Tanta University, Tanta 31527, Egypt; 5Ophthalmology Department, Kafr Ash Shaykh Ophthalmology Hospital, Kafr Ash Shaykh 33511, Egypt; 6McLaren Health Care, Flint, MI 48532, USA; 7Internal Medicine Department, Michigan State University, East Lansing, Michigan, MI 48824, USA; 8Harvard Medical School, Postgraduate Medical Education, Boston, MA 02115, USA; 9Doheny Eye Institute, University of California, Los Angeles, CA 90033, USA

**Keywords:** COVID-19, vaccine, uveitis, vaccine-associated uveitis

## Abstract

Numerous complications following COVID-19 vaccination has been reported in the literature, with an increasing body of evidence reporting vaccination-associated uveitis (VAU). In this systematic review, we searched six electronic databases for articles reporting the occurrence of VAU following COVID-19 vaccination. Data were synthesized with emphasis on patients’ characteristics [age, gender], vaccination characteristics [type, dose], and outcome findings [type, nature, laterality, course, location, onset, underlying cause, and associated findings]. Data are presented as numbers (percentages) for categorical data and as mean (standard deviation) for continuous data. Sixty-five studies were finally included [43 case reports, 16 case series, four cohort, one cross-sectional, and one registry-based study]. VAU occurred in 1526 cases, most commonly in females (68.93%) and middle-aged individuals (41–50 years: 19.71%), following the first dose (49.35%) of vaccination, especially in those who received Pfizer (77.90%). VAU occurred acutely (71.77%) as an inflammatory reaction (88.29%) in unilateral eyes (77.69%), particularly in the anterior portion of the uvea (54.13%). Importantly, most cases had a new onset (69.92%) while only a limited portion of cases had a reactivation of previous uveitis condition. In conclusion, although rare, uveitis following COVID-19 vaccination should be considered in new-onset and recurrent cases presenting with either acute or chronic events.

## 1. Introduction

Following the declaration of the COVID-19 pandemic, healthcare systems all over the globe were burdened by the increased number of daily diagnosed cases and associated deaths. This urged the need to develop effective vaccines within a short time period. During the past two years, a number of vaccines received emergency authorization and were then disseminated globally. Currently, over 12.6 billion doses have been distributed, including messenger RNA vaccines (the Pfizer-BioNTech and the Moderna), vector vaccines (Johnson & Johnson, AstraZeneca), and protein subunit vaccine (Novavax). Although these vaccines were effective in limiting the spread of the disease and limiting the occurrence of severe forms, several adverse events were reported, particularly those involving the eye [1,2].

Uveitis is a potentially vision-threatening condition that involves intraocular inflammation. It accounts for 10–15% of blindness cases worldwide [3]. It has incidence and prevalence rates of 50.45 and 9–730 per 100,000 cases, respectively. The etiology of uveitis is multifactorial, which can be autoimmune, systemic (60.1%), infectious (30–50%), or idiopathic (20–40%) in origin [3]. That being said, uveitis can occur postvaccination. Vaccination-related uveitis, although uncommon, has been reported during previous Hepatitis B virus (40%), human papillomavirus (15%), hepatitis A virus, influenza virus, Bacille-Calmette-Guerin, varicella virus, and measles-mumps-rubella vaccination programs [4]. In this context, the hypothesis of COVID-19 vaccine-associated uveitis (VAU) has emerged, with several reports highlighting the magnitude of this problem [1,5,6]. For instance, non-infectious uveitis has been reported in 66.8 and 62.7 cases per 100,000 person-years following the first and second doses of the BNT162b2 mRNA vaccine, respectively [7].

In light of COVID-19 VAU, the level of evidence has shifted. Instead of being based solely on case reports, a number of new cohort and cross-sectional studies have been published in this regard, all of which highlight the magnitude of this newly emerging observation. Therefore, we conducted a systematic review to summarize available evidence on COVID-19 vaccine-associated uveitis with a particular focus on vaccines’ information [type, dose, duration], patients’ characteristics [sociodemographic and clinical], and disease-associated outcomes [origin, type, location, presentation, management, and outcomes].

## 2. Materials and Methods

### 2.1. Study Protocol and Database Search

This research was carried out in accordance with the Preferred Reporting for Systematic Review and Meta-Analysis (PRISMA) recommendations. In July 2022, our protocol was registered on PROSPERO [registration number: CRD42022358117]. Meanwhile, on 26–27 July 2022, we searched six electronic databases [PubMed, Scopus, EMBASE, Web of Science, CENTRAL, and Google Scholar] to retrieve all studies that reported the occurrence of VAU following COVID-19 vaccination using the following keywords: COVID-19 AND vaccin* AND uveitis. Medical Subject Headings (MESH) terms were also added whenever applicable to retrieve all relevant studies based on their indexed terms in included databases. Of note, only the first 200 records from Google Scholar were retrieved and screened as per the recent recommendations [8]. The detailed search strategy for each database is provided in Table A1. It is worth noting that an updated database search was carried out on 11 September 2022, to include any newly published studies before the official synthesis of collected data.

Additionally, on 1 September 2022, after finishing the screening process, we conducted a manual search of references to identify any relevant studies that we could not identify through the original database search. This search was conducted through: (1) the reference list of included studies, (2) “similar article” of included papers on PubMed, and (3) Google by using these keywords: “COVID-19” + “uveitis” + “vaccine”.

### 2.2. Eligibility Criteria

We included original research papers that discussed the occurrence of VAU following COVID-19 vaccination. We included all of the following study designs: case reports and series, cohorts, and cross-sectional studies. Studies were included regardless of the type and/or dose of vaccine, history or location of uveitis, or the language of publication. Meanwhile, studies were excluded if they: (1) recruited individuals (healthy or infected with COVID-19) who did not receive COVID-19 vaccines, (2) were not original (reviews, editorials, commentaries, books, etc.), (3) included duplicated records, (4) did not have a full text, or (5) reported other types of ocular/ophthalmic complications other than uveitis.

### 2.3. Screening and Study Selection

Retrieved records from the database search were exported into EndNote software for duplicate removal before the beginning of the screening phase. Records were then imported into an Excel (Microsoft, Rochester, MN, USA) Sheet for screening. The screening was divided into two steps: title and abstract screening and full-text screening. The full texts of eligible articles were then retrieved for screening before being finally included in the review. Both steps were carried out by three reviewers (HY, HA, AE). Any differences between reviewers were solved through group discussions, and the senior authors (YYSC, CD) were consulted if reviewers could not reach an agreement.

### 2.4. Data Extraction

The data extraction was performed by three reviewers [HY, HA, AE] through a data extraction sheet that was formatted through Excel (Microsoft, Rochester, MN, USA). This sheet consisted of six parts. The first part included the baseline characteristics of included studies (title, authors’ names, year of publication, country, and study design) and patients as well [sample size, age, and gender]—only those with evidence VAU. The second part included data on the administered vaccines (type and dose), while the third part included information regarding VAU cases’ medical history (systemic, immunological, and ocular diseases). The fourth part included patients’ clinical presentation and examination findings (symptoms, signs, and intraocular pressure (IOP)), while the fifth part included data on the main outcome of interest—VAU (type, location, laterality, interval between vaccination and symptom onset, nature, underlying cause, and associated findings). The final part included the management approach in such cases and reported outcomes (resolution, improvement, complications, recurrence, etc.).

### 2.5. Data Synthesis

All of the included studies were qualitatively analyzed as per our plan in priori. Additionally, since data were provided on a per-case basis, we performed several descriptive analyses to detect patterns on the occurrence of VAU based on: age, gender, type and dose of vaccination, presenting symptoms, laterality (right, left, unilateral, bilateral), type of uveitis (new-onset, reactivation), longevity (acute, chronic), location (anterior, intermediate, posterior, panuveitis), duration of vaccination to symptom onset, associated findings [macular edema, glaucoma, synechiae], and management outcomes (resolution, improvement, recurrence, complications). Data are presented as mean/standard deviation (SD) for continuous variables and as numbers/percentages for categorical ones.

## 3. Results

### 3.1. Search Results

The results of the database search and screening phases are presented in Figure 1. The initial database search yielded 538 articles, of which 209 duplicates were removed through EndNote. Following the screening of 329 articles, the full texts of 65 articles were retrieved for full-text screening, of which 11 articles were excluded. The manual search resulted in 10 articles, and an updated database search revealed one additional article, resulting in a total number of relevant eligible studies of 65.

### 3.2. Baseline Characteristics of Studies Reporting COVID-19 VAU

A total of 65 studies were both qualitatively and quantitatively analyzed (Table 1) [1,5,7,9,10,11,12,13,14,15,16,17,18,19,20,21,22,23,24,25,26,27,28,29,30,31,32,33,34,35,36,37,38,39,40,41,42,43,44,45,46,47,48,49,50,51,52,53,54,55,56,57,58,59,60,61,62,63,64,65,66,67,68,69], out of which 43 were case reports, 16 were case series, four were retrospective cohort, one was cross-sectional, and one was a registry-based study. The sample size of included patients with VAU ranged from 1 to 1094, with an overall sample size of 1526 VAU cases. Most reports were from India (*n* = 9), followed by China (*n* = 7), Israel (*n* = 7), Korea (*n* = 6), Italy (*n* = 4), and USA (*n* = 4), respectively.

### 3.3. Sociodemographic and Clinical Characteristics of VAU Cases

#### 3.3.1. Age and Gender

COVID-19 vaccine-associated uveitis was twice more likely to occur in females than in males (68.93% vs. 31.06%). Patients’ age ranged from 8 to 95 years (mean of 48.18 and standard deviation of 18.94). The peak of vaccine-associated uveitis occurred in middle-aged patients (41–50 years of age) with a declining trend as it comes nearer to both extremes (0–10 or ≥91 years) [Table 2].

#### 3.3.2. Medical History

A minority of VAU cases reported having either systemic, ocular, or immunological diseases [Table 3]. SARS-CoV-2 infection occurred in a very limited number of patients (0.93%), while only 0.59% and 0.42% reported having hypertension or diabetes, respectively. More than one-tenth of VAU cases reported having uveitis in the past (13.51%). Although cases reported a wide variety of previous ocular conditions, none of them seems to be correlated with VAU due to their rare occurrence (below 0.50%). In terms of immunological diseases, autoimmune diseases (AIDS) were the most frequent among VAU cases, accounting for 1.19% of cases.

### 3.4. Vaccine- and Outcome-Related Characteristics

#### 3.4.1. Type and Dose of COVID-19 Vaccines

The majority of cases were documented in those who took the Pfizer-BioNTech vaccine (77.9%), followed by Moderna (15.54%), and AstraZeneca (3.01%), respectively [Table 2]. Most cases were more likely to occur following the first dose of the vaccine (49.35%), and the occurrence of vaccine-related uveitis was remarkedly minimized following the third and fourth booster doses (7.84% and 0.37%), respectively.

#### 3.4.2. Clinical Presentation

The majority of patients presented with redness (72.99%), followed by diminished vision (23.53%), photophobia (10.48%), and blurry vision (5.28%), respectively [Table 4]. Although infrequent, some VAU cases presented with floaters (2.22%) and vision loss (1.07%). The IOP was measured in 38 and 17 right and left eyes of VAU cases, respectively. IOP ranged from 9 to 55 and from 8 to 60 mmHg in the left and right eyes, with a mean IOP of 17.93 (SD = 11.03) and 17.3 (SD = 9.38) mmHg, respectively.

#### 3.4.3. The Nature of the Reported VAU and Disease Laterality

Out of 1526 VAU cases, only 1476 cases had the type of intraocular inflammation documented [Table 5]. The most common type was uveitis (97.56%), followed by VKH (1.08%) and retinochoroiditis (0.20%), respectively. The mean interval from COVID-19 vaccination to the occurrence of uveitis was 9.61 (SD = 8.07) days, ranging from 1 to 42 days post-vaccination. VAU was twice more likely to occur in one eye/unilaterally (77.69%) than in both eyes/bilaterally (22.05%). The rate of VAU occurrence in the right and left eyes was comparable (32.3% vs. 34.61%), respectively [Table 5].

#### 3.4.4. Disease Course, Location, Nature, and Underlying Cause of VAU

The course of VAU was acute in more than two-thirds of the population (71.77%) as compared to chronic cases (28.22%) [Table 5]. Among VAU cases where the location was determined, the anterior segment of the uveal tract in more than half the population was affected (54.13%). Surprisingly, panuveitis was more likely to occur than posterior uveitis by almost two-fold (10.02% vs. 5.28%). Of note, the majority of VAU cases did not have a history of uveitis and experienced an episode of uveitis for the first time (69.92%), while only one-third of VAU cases had prior episodes of uveitis (30.08%). VAU was inflammatory in nature in most cases (88.29%) and infectious in 8.36% of them. The underlying cause of VAU was idiopathic in almost half the population (43.26%), while VKH (7.34%) accounted for the most commonly reported cause among other causes [Table 5].

### 3.5. Management and Treatment Outcomes

A detailed description of the management plan that was carried out per each patient is provided in Table A2. Of note, the majority (90.15%) of VAU cases showed complete resolution following treatment, while only 9.85% had partial improvement. In studies that assessed complications following the treatment of VAU cases, 21.68% of cases had at least one complication, the most common of which being transient elevation in IOP (non-serious) and nummular corneal lesions in 3.61% of cases [Table 6].

## 4. Discussion

Since the emergence of COVID-19 vaccines, many adverse events have been recognized globally. Of these adverse events, uveitis was one of the most commonly reported ocular events. Specifically, a recent study, using the CDC-VAERS registry, reported that VAU was evident in 1094 VAU cases across 40 countries with a crude incidence rate of 0.57 cases per million doses of the COVID-19 Pfizer vaccine [71].

The exact pathophysiology of VAU is unclear, but it is believed to be mediated through an autoimmune reaction by the vaccines [72]. Additionally, it could be due to a combination of mechanisms such as molecular mimicry, the production of specific autoantibodies, hypersensitivity reactions, and the role of some vaccine adjuvants [73,74]. Vaccines provoke an inflammatory cascade by expression of type 1 interferon, resulting in a host immune response. On the other hand, they may also induce the production of autoantibodies, which can potentially trigger an autoimmune reaction [73]. Rabinovitch et al. [55] suggested that VAU caused by mRNA vaccines is a type I autoimmune reaction resulting in spiked levels of type 1 interferon. Cunningham et al. [75] attributed VAU to type 4 hypersensitivity reaction due to molecular mimicry between uveal self-peptides and vaccine peptides. Nevertheless, the postulated mechanisms that lead to VAU following COVID-19 vaccinations are mainly hypothetical and warrant additional studies. Due to the autoimmunity nature of VAU, it tends to occur more frequently in females [76]. Although the underlying cause of this trend is uncertain, the latest evidence has shown that sex hormones have an impact on the immune reaction, with estrogen enhancing it and androgens repressing it [77]. Furthermore, recent research has demonstrated that estrogen is essential for the development and function of Th17 cells in addition to IL-17 generation [78]. Our findings coincide with this, showing that COVID-19 VAU was twice more likely to occur in females than in males (68.93% vs. 31.06%). In addition, AIDS was the most frequent among VAU cases, accounting for 1.19% of cases, which strengthens the hypothesis of autoimmunity.

Our study supports the hypothesis that uveitis can occur following COVID-19 vaccination either as new-onset (the majority of cases) or as an exacerbation or reactivation of a previous uveitis. The peak of vaccine-associated uveitis occurred in middle-aged individuals (41–50 years of age), which is parallel to findings made by Darrell et al. [79]. The majority of cases were documented in those who took the Pfizer-BioNTech vaccine (77.9%), followed by Moderna (15.54%), and AstraZeneca (3.01%), respectively. This could be explained by the fact that Pfizer–BioNTech COVID-19 vaccine elicits an additional CD8 T-cell immune response, providing additional protection against SARS-CoV2 infection—however, also triggering autoimmune reactions [80,81]. This could also be attributed to the dominance of Pfizer-BioNTech vaccine over other COVID-19 vaccine type in the number of administered doses. For instance, up to December 2022, 656.90 million Pfizer doses have been administered followed by Moderna (153.82 million), AstraZeneca (67.03 million), Jhonshon&Jhonson (18.93 million), Sinopharm (2.32 million), and Sputnik (1.85 million), respectively. Other vaccines (Sinovac, Novavax, and Covaxin) have been administered at a much lower rate (below 1 million doses) [82]. These findings were also observed in Singh RB et al. [71]’s registry analysis. Most cases were more likely to occur following the first dose of the vaccine (49.35%), and the occurrence of vaccine-related uveitis was remarkedly minimized following the third and fourth booster doses (7.84% and 0.37%), respectively. Oberhardt V et al. [83] found that the first dose of COVID-19 vaccines is associated with inducing a significantly higher level of anti-spike IgG protein, resulting in a proportionately higher number of naive and transitional B cells, as well as functional spike-specific CD8+ T cells, which is parallel to findings observed by Singh RB et al. [71]. The mean interval from COVID-19 vaccination till to the occurrence of uveitis was 9.61 (SD = 8.07) days. This might be explained by the fact that the highest immune response usually occurs during the first ten days [84]. Unfortunately, given the small sample size, we could not determine the interval time from vaccination to symptom onset per each vaccine type. That being said, the previous study [71] indicated that the interval is significantly longer in those who received the Moderna vaccine as compared to those who received either Pfizer or AstraZeneca (*p* < 0.0001). However, no conclusive, clinically applicable evidence can be drawn from such observations given the non-normal distribution of analyzed data (standard deviation was larger than the mean).

Moreover, the occurrence of VAU following COVID-19 vaccination did not follow a specific pattern regarding the location of uveitis or the course of the disease. However, in our study, VAU was more likely to occur as an acute inflammatory (non-infectious) reaction involving mainly the anterior portion of the uveal tract. The majority of VAU cases did not have a history of uveitis and experienced an episode of uveitis for the first time (69.92%), while only one-third of VAU cases had prior episodes of uveitis (30.08%). This necessitates the importance of early recognition of symptoms of uveal tract involvement especially diminished vision, photophobia, and blurry vision. Similar to any uveitis, the management of COVID-19 VAU is an exclusion diagnosis. Therefore, identifying an underlying cause whilst also ruling out infections is critical. In addition to the standard uveitis questionnaires for previous uveitis, medical history, and constitutional health symptoms during the COVID-19 pandemic, clinicians should inquire about COVID-19 vaccination status.

## 5. Conclusions

Our review summarizes the occurrence of COVID-19 vaccination-associated uveitis, which is more likely to occur among middle-aged females. This event occurs either as a new onset of the disease or a reactivation of previous uveitis, most commonly after vaccination with Pfizer vaccine. Although it commonly occurs after the first and then second doses of the vaccines, it can occur after the first and second booster doses as well. It usually involves the anterior part of the uveal tract as an inflammatory event in an acute form.

## Figures and Tables

**Figure 1 vaccines-11-00069-f001:**
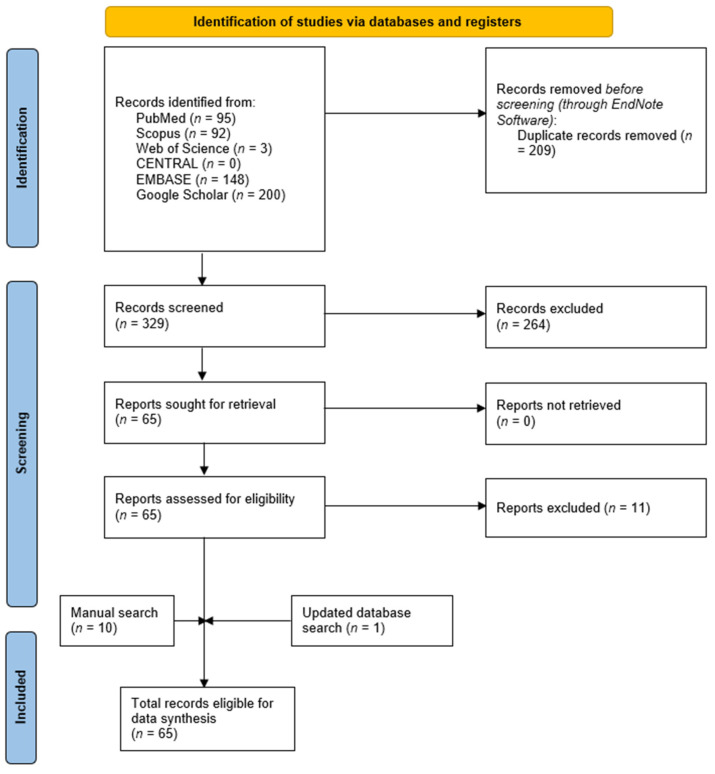
A PRISMA flow diagram showing the database search and screening results.

**Table 1 vaccines-11-00069-t001:** The baseline characteristics of studies reporting COVID-19 vaccine-associated uveitis cases.

Author (YOP)	Country	Design	Gender	Age	Type	Dose
Case Reports						
Accorinti (2022) [9]	Italy	Case Report	Female	54	Pfizer-BioNTech	First
Achiron (2022) [10]	Finland	Case Report	Male	17	Pfizer-BioNTech	-
Al-Allaf (2022) [11]	Qatar	Case Report	Male	46	Pfizer-BioNTech	First
Alhamazani (2022) [12]	Saudi Arabia	Case Report	Male	37	Pfizer-BioNTech	First
Brunet de Courssou (2022) [16]	France	Case Report	Female	57	Pfizer-BioNTech	First
Chen (2022a) [18]	China	Case Report	Male	19	Sinovac	First
De Carvalho (2022) [22]	Brazil	Case Report	Male	51	Oxford/AstraZeneca	First
De Domingo (2022) [23]	Spain	Case Report	Female	46	Pfizer-BioNTech	Second
Ishay (2021) [32]	Israel	Case Report	Male	28	Pfizer-BioNTech	First
Duran (2022) [25]	Israel	Case Report	Female	54	Pfizer-BioNTech	First
ElSheikh (2021) [26]	Egypt	Case Report	Female	18	Sinopharm	First
Lee (2022) [39]	South Korea	Case Report	Female	83	Pfizer-BioNTech	Second
Gedik (2022) [28]	Turkey	Case Report	Male	47	Pfizer-BioNTech	Second
Goyal (2021) [29]	India	Case Report	Male	34	Oxford/AstraZeneca	Second
Hébert (2022) [30]	Canada	Case Report	Male	41	Pfizer-BioNTech	First
Hwang (2022) [31]	South Korea	Case Report	Female	21	Pfizer-BioNTech	Second
Jain (2021) [33]	India	Case Report	Male	27	Oxford/AstraZeneca	First
Joo (2022) [34]	South Korea	Case Report	Female	50	Moderna	First
Kim (2022) [36]	South Korea	Case Report	Female	72	Oxford/AstraZeneca	First
Koong (2021) [37]	Singapore	Case Report	Male	54	Pfizer-BioNTech	First
Papasavvas (2021) [53]	Switzerland	Case Report	Female	73	Moderna	First
Ding (2022) [24]	China	Case Report	Male	33	Sinopharm	First
Lee (2022) [40]	USA	Case Report	Female	25	Moderna	Second
Mahendradas (2022) [42]	India	Case Report	Female	19	Covaxin	Second
Matsuo (2022) [43]	Japan	Case Report	Male	34	Pfizer-BioNTech	Second
Mishra (2021) [44]	India	Case Report	Male	71	Oxford/AstraZeneca	First
Mudie (2021) [45]	Spain	Case Report	Female	43	Pfizer-BioNTech	Second
Pan (2021) [51]	China	Case Report	Female	50	Sinopharm	
Papasavvas (2021) [53]	Switzerland	Case Report	Female	43	Pfizer-BioNTech	Second
Reddy (2021) [57]	India	Case Report	Female	30	Oxford/AstraZeneca	Second
Renisi (2021) [59]	Italy	Case Report	Male	23	Pfizer-BioNTech	Second
Sangoram (2022) [60]	India	Case Report	Female	40	Oxford/AstraZeneca	Second
Santiago (2021) [61]	Puerto Rico	Case Report	Male	32	Pfizer-BioNTech	Second
Saraceno (2021) [62]	Brazil	Case Report	Female	62	Oxford/AstraZeneca	-
Singh (2022a) [64]	India	Case Report	Male	29	Oxford/AstraZeneca	First
Yalçinkaya (2022) [67]	Turkey	Case Report	Male	12	Pfizer-BioNTech	First
Yamaguchi (2022) [68]	Japan	Case Report	Female	30	Pfizer-BioNTech	Second
Shilo (2022) [70]	Israel	Case Report	Male	20	Pfizer-BioNTech	First
Kakarla (2022) [35]	India	Case Report	Female	15	Covaxin	First
Numakura (2022) [48]	Japan	Case Report	Male	61	Pfizer-BioNTech	First
Murgova (2022) [46]	Bulgaria	Case Report	Female	89	Pfizer-BioNTech	Second
Patel (2022) [54]	USA	Case Report	Male	79	Pfizer-BioNTech	Second
Lawson-Tovey (2022) [38]	Italy	Case Report	Female	-	-	Second
**Case Series**						
**Author (YOP)**	**Country**	**Cases**	**Gender**	**Age**	**Type**	**Dose**
Arora (2022) [13]	India	Case 1	Female	20	Oxford/AstraZeneca	First
		Case 2	Male	26	Oxford/AstraZeneca	First
Choi (2022) [20]	Korea	Case 1	Male	62	Oxford/AstraZeneca	First
		Case 2	Female	79	Pfizer-BioNTech	First
		Case 3	Female	55	Pfizer-BioNTech	First
Ortiz-Egea (2022) [49]	Spain	Case 1	Female	92	Pfizer-BioNTech	First
		Case 2	Female	85	Pfizer-BioNTech	First
Nanji (2022) [47]	USA	Case 1	Female	58	Moderna	First
		Case 2	Female	60	Moderna	First
Pang (2022) [52]	China	Case 1	Female	50	Sinopharm	First
		Case 2	Female	34	Sinopharm	Second
Ren (2022) [58]	China	Case 1	Female	46	Sinovac	First
		Case 2	Female	26	Sinovac	First
Cohen (2022) [21]	Israel	Case 1	Female	81	Pfizer-BioNTech	Second
		Case 2	Female	64	Pfizer-BioNTech	Second
		Case 3	Male	74	Pfizer-BioNTech	Third
		Case 4	Male	63	Pfizer-BioNTech	Third
Aguiar (2022) [17]	Portugal	Case 1	Female	21	Pfizer-BioNTech	First
		Case 2	Male	70	Pfizer-BioNTech	Second
De Queiroz Tavares Ferreira (2022) [5]	Brazil	Case 1	Female	27	Oxford/AstraZeneca	First
		Case 2	Male	39	Oxford/AstraZeneca	First
		Case 3	Female	38	Pfizer-BioNTech	First
		Case 4	Female	32	Sinovac	Second
Chen (2022b) [19]	China	Case 1	Male	33	Sinopharm	First
		Case 2	Female	57	Sinovac	Second
		Case 3	Male	21	Sinovac	First
		Case 4	Female	30	Sinovac	Second
		Case 5	Female	28	Sinopharm	First
Chew (2022) [1]	Singapore	Case 1	Female	64	Pfizer-BioNTech	Second
		Case 2	Male	74	Sinopharm	Second
		Case 3	Female	31	Pfizer-BioNTech	Second
		Case 4	Female	71	Pfizer-BioNTech	Second
		Case 5	Female	32	Pfizer-BioNTech	Second
		Case 6	Female	28	Pfizer-BioNTech	Second
Rallis (2022) [56]	UK	Case 1	Female	47	Oxford/AstraZeneca	First
		Case 2	Female	48	Oxford/AstraZeneca	First
		Case 3	Male	44	Oxford/AstraZeneca	First
		Case 4	Female	59	Oxford/AstraZeneca	First
		Case 5	Male	65	Oxford/AstraZeneca	First
		Case 6	Male	95	Pfizer-BioNTech	First
		Case 7	Male	68	Pfizer-BioNTech	First
Li (2022) [41]	China	Case 1	Female	48	Sinovac	First
		Case 2	Male	41	Sinovac	Third
		Case 3	Male	8	Sinovac	First
		Case 4	Female	52	Sinovac	Second
		Case 5	Female	55	Sinovac	First
		Case 6	Female	67	Sinovac	First
		Case 7	Female	46	Sinovac	First
		Case 8	Female	57	Sinovac	First
		Case 9	Male	22	Sinovac	First
Sim (2022) [63]	Korea	Case 1	Female	51	Pfizer-BioNTech	Second
		Case 2	Female	21	Pfizer-BioNTech	Second
		Case 3	Male	50	Pfizer-BioNTech	Second
		Case 4	Female	52	Pfizer-BioNTech	Third
		Case 5	Male	32	Johnson & Johnson	Second
		Case 6	Male	72	Oxford/AstraZeneca	Second
		Case 7	Female	67	Oxford/AstraZeneca	Third
		Case 8	Male	54	Pfizer-BioNTech	Second
		Case 9	Female	61	Pfizer-BioNTech	Third
		Case 10	Female	63	Pfizer-BioNTech	Second
		Case 11	Female	47	Pfizer-BioNTech	Third
Rabinovitch (2021) [55]	Israel	Case 1	Female	43	Pfizer-BioNTech	First
		Case 2	Male	34	Pfizer-BioNTech	First
		Case 3	Female	34	Pfizer-BioNTech	First
		Case 4	Male	78	Pfizer-BioNTech	Second
		Case 5	Male	53	Pfizer-BioNTech	First
		Case 6	Male	64	Pfizer-BioNTech	First
		Case 7	Male	68	Pfizer-BioNTech	First
		Case 8	Female	61	Pfizer-BioNTech	First
		Case 9	Male	59	Pfizer-BioNTech	Second
		Case 10	Male	72	Pfizer-BioNTech	Second
		Case 11	Male	51	Pfizer-BioNTech	Second
		Case 12	Female	42	Pfizer-BioNTech	Second
		Case 13	Male	74	Pfizer-BioNTech	Second
		Case 14	Male	39	Pfizer-BioNTech	Second
		Case 15	Female	64	Pfizer-BioNTech	Second
		Case 16	Female	50	Pfizer-BioNTech	Second
		Case 17	Female	23	Pfizer-BioNTech	Second
		Case 18	Female	65	Pfizer-BioNTech	First
		Case 19	Male	36	Pfizer-BioNTech	Second
		Case 20	Male	41	Pfizer-BioNTech	Second
		Case 21	Female	28	Pfizer-BioNTech	Second
Bolletta (2021) [15]	Italy	Case 1	Male	79	Oxford/AstraZeneca	Second
		Case 2	Female	65	Pfizer-BioNTech	Second
		Case 3	Female	42	Oxford/AstraZeneca	Second
		Case 4	Female	52	Pfizer-BioNTech	Second
		Case 5	Male	44	Pfizer-BioNTech	First
		Case 6	Female	35	Moderna	Second
		Case 7	Male	47	Pfizer-BioNTech	First
		Case 8	Female	58	Pfizer-BioNTech	First
		Case 9	Female	52	Oxford/AstraZeneca	First
		Case 10	Female	44	Pfizer-BioNTech	Second
		Case 11	Female	58	Pfizer-BioNTech	Second
		Case 12	Female	47	Pfizer-BioNTech	First
		Case 13	Female	68	Pfizer-BioNTech	Second
**Observational Studies**						
**Author (YOP)**	**Country**	**Design**	**Male/Total**	**Age** **Mean (SD)**	**Type [N]**	**Dose [N]**
Ferrand (2022) [27]	Germany	Retrospective Cohort	6/25	43.2 (13.9)	Pfizer-BioNTech [15]; Oxford/AstraZeneca [6]; Moderna [3]; Covaxin [1]	First [6]; Second [19]
Ozdede (2022) [50]	Istanbul	Cross-Sectional	-/5	-	Sinovac [3]; Pfizer-BioNTech [2]	-
Testi (2022)	UK	Retrospective Cohort	22/50	41.3 (13.9)	Pfizer-BioNTech [24]; Oxford/AstraZeneca [16]; Moderna [8]; Sinopharm [1]; Covaxin [1]	First [28]; Second [22]
Tomkins-Netzer (2022) [7]	Israel	Retrospective Cohort	-	-	Pfizer-BioNTech [188]	First [100]; Second [88]
Barda (2021) [14]	Israel	Retrospective Cohort	-/26	-	Pfizer-BioNTech [26]	-
Singh (2022b) [71]	USA	Retrospective registry-based	322/1094	46.24 (16.93)	Pfizer-BioNTech [853]; Moderna [220]; Johnson & Johnson [21]	First [452]; Second [373]; Third [97]; Fourth [5]

YOP: Year of Publication; UK: United Kingdom; USA: United States of America; N: Number of Cases; SD: Standard Deviation.

**Table 2 vaccines-11-00069-t002:** Patients’ and COVID-19 vaccines’ characteristics.

Variable	Subgroup	Number	%
**Gender**			
	Male	406	31.06
	Female	901	68.93
**Age**			
	1–10	1	0.73
	11–20	8	5.84
	21–30	20	14.6
	31–40	19	13.87
	41–50	27	19.71
	51–60	23	16.79
	61–70	20	14.6
	71–80	13	9.49
	81–90	4	2.92
	≥91	2	1.46
**Vaccine Type**			
	Covaxin	4	0.26
	Johnson & Johnson	22	1.44
	Moderna	237	15.54
	Oxford/AstraZeneca	46	3.01
	Pfizer-BioNTech	1188	77.9
	Sinovac	11	0.72
	Sinopharm	17	1.11
**Vaccine Dose**			
	First	654	49.35
	Second	562	42.41
	Third	104	7.84
	Fourth	5	0.37

**Table 3 vaccines-11-00069-t003:** The systemic, ocular, and immunologic history of vaccine-associated uveitis cases.

Author (YOP)	Medical History			Total
	Systemic [N]	Ocular [N]	Immunological [N]	
Accorinti (2022)	None	-	-	1
Achiron (2022)	-	Uveitis [1]—RVO [1]—Iridis Rubeosis [1]	-	1
Al-Allaf (2022)	HTN	-	-	1
Alhamazani (2022)	-	-	-	1
Brunet de Courssou (2022)	-	-	-	1
Chen (2022a)	None	None	None	1
De Carvalho (2022)	AS	Uveitis [1]	HLA-B27 [1]	1
De Domingo (2022)	-	None	None	1
Ishay (2021)	Bechet’s disease	None	Bechet’s disease [1]	1
Duran (2022)	DM	None	-	1
ElSheikh (2021)	-	None	JIA [1]	1
Lee (2022)	HTN—Lipidemia	None	-	1
Gedik (2022)	-	-	-	1
Goyal (2021)	None	None	None	1
Hébert (2022)	None	None	None	1
Hwang (2022)	None	None	None	1
Jain (2021)	-	Uveitis [1]	JIA [1]	1
K. Joo (2022)	None	Allergic conjunctivitis [1]	None	1
Kim (2022)	-	-	-	1
Koong (2021)	DM—Lipidemia	None	None	1
Papasavvas (2021)	None	Cataract [1]	None	1
Ding (2022)	HTN	None	None	1
Lee (2022)	-	-	None	1
Sai (2022)	-	Uveitis [1]	JIA [1]	1
Matsuo (2022)	-	-	-	1
Mishra (2021)	DM—HTN	-	-	1
Mudie (2021)	-	-	-	1
Pan (2021)	-	-	-	1
Papasavvas (2021)	-	VKH [1]	None	1
Reddy (2021)	-	-	-	1
Renisi (2021)	None	None	None	1
Sangoram (2022)	None	None	None	1
Santiago (2021)	None	None	None	1
Saraceno (2021)	None	None	None	1
Singh (2022a)	None	None	None	1
Yalçinkaya (2022)	None	None	None	1
Yamaguchi (2022)	None	None	None	1
Shilo (2022)	None	None	None	1
Kakarla (2022)	None	None	None	1
Numakura (2022)	None	None	None	1
Murgova (2022)	-	CRVO [1]—Cataract [1]—Glaucoma [1]—Herpetic uveitis [1]	-	1
Patel (2022)	None	Cataract [1]—RD [1]—ERM [1]	None	1
Lawson-Tovey (2022)	-	-	-	1
Arora (2022)	-	Uveitis [2]—SLC [1]	-	2
Choi (2022)	HTN [2]—DM [1]—Asthma [1]	Uveitis [2]—BRVO [1]	HLAB51 [1]	3
Ortiz-Egea (2022)	-	AMD [1]	-	2
Nanji (2022)	-	Uveitis [1]—OU [1]	-	2
Pang (2022)	-	-	-	2
Ren (2022)	-	-	-	2
Cohen (2022)	None	HZO [1]—Uveitis [1]	Psoriasis [1]—RA [1]	4
Aguiar (2022)	Epilepsy [1]—Asthma [1]—DM [1]—HTN [1]—Rhinitis [1]	None	None	2
Ferreira (2022)	COVID-19 [2]—HTN [1]	None	None	4
Chen (2022b)	AS [1]	-	-	5
Chew (2022)	None	Uveitis [3]—Cataract [1]—PACG [1]—HSK [4]	HLAB51 [1]	6
Rallis (2022)	-	-	-	7
Li (2022)	-	-	-	9
Sim (2022)	-	-	-	11
Rabinovitch (2021)	AS [3]—Psoriasis [2]—Crohn’s disease [1]—Spondylarthritis [1]	Uveitis [8]—HZO [1]	-	21
Bolletta (2021)	Spondylarthritis [1]—Psoriatic arthritis [1]	Uveitis [3]—VKH [2]—Toxoplasma Retinochoroiditis [2]	-	13
Ferrand (2022)	-	Uveitis [19]—VKH [1]	HLAB27 [2]—MS [2]—JIA [1]	25
Ozdede (2022)	-	-	Bechet’s syndrome [1]	5
Testi (2022)	-	Uveitis [20]—Glaucomatocyclitic Crisis [3]		50
Tomkins-Netzer (2022)	-	-	-	188
Barda (2021)	-	-	-	26
Singh (2022b)	COVID-19 [9]	Uveitis [106]	AIDs [14]	1094
Summary of the History of VAU Cases				
**Category**	**Disease**	**Number**	**Total**	**%**
**Systemic Diseases**				
	HTN	7	1178	0.59
	DM	5	1178	0.42
	AS	5	1178	0.42
	Lipidemia	2	1178	0.16
	Asthma	2	1178	0.16
	Epilepsy	1	1178	0.08
	Rhinitis	1	1178	0.08
	COVID-19	11	1178	0.93
**Ocular Diseases**				
	Uveitis	170	1258	13.51
	VKH	4	1258	0.32
	HZO	2	1258	0.16
	Toxoplasma Retinochoroiditis	2	1258	0.16
	Glaucoma	5	1258	0.39
	Cataract	4	1258	0.32
	HSK	4	1258	0.32
	SLC	1	1258	0.07
	BRVO	1	1258	0.07
	RVO	1	1258	0.07
	Iridis Rubeosis	1	1258	0.07
	OU	1	1258	0.07
	CRVO	1	1258	0.07
	ERM	1	1258	0.07
	RD	1	1258	0.07
	Conjunctivitis	1	1258	0.07
	AMD	1	1258	0.07
**Immunological Diseases**				
	HLA-B27	3	1170	0.26
	JIA	4	1170	0.34
	Psoriasis	3	1170	0.26
	HLAB51	2	1170	0.17
	MS	2	1170	0.17
	Bechet’s disease	3	1170	0.26
	AIDs	14	1170	1.19
	RA	1	1170	0.08
	Crohn’s disease	2	1170	0.17
	Spondylarthritis	1	1170	0.08

AS: Ankylosing Spondylitis; AMD: Age-related Macular Degeneration; BRVO: Branch Retinal Vein Occlusion; CRVO: Central Retinal Vein Occlusion; ERM: Epiretinal Membrane; HTN: Hypertension; HSK: Herpes-Simplex Keratitis; HZO: Herpes-Zoster Ophthalmicus; JIA: Juvenile Idiopathy Arthritis; OU: Optic Disc Vasculitis; PACG: Primary Angle-Closure Glaucoma; RD: Retinal Detachment; SLC: Serpiginous-like Choroiditis; VAU: Vaccine-Associated Uveitis; VKH: Vogt-Koyanagi-Harada.

**Table 4 vaccines-11-00069-t004:** The clinical presentation of COVID-19 vaccine-associated uveitis.

Author (YOP)	Eye Symptoms/Signs [N]	Total	
Accorinti (2022)	Central scotoma [1]	1	
Achiron (2022)	Vision loss [1]	1	
Al-Allaf (2022)	Pain [1]—Erythema [1]—Photophobia [1]—Blurry vision [1]	1	
Alhamazani (2022)	Pain [1]—Photophobia [1]—Redness [1]—Diminished Vision [1]	1	
Brunet de Courssou (2022)	Headache [1]—Blurry vision [1]	1	
Chen (2022a)	Headache [1]—Blurry vision [1]—Fatigue [1]	1	
De Carvalho (2022)	Pain [1]—Redness [1]—Foreign Body Sensation [1]	1	
De Domingo (2022)	Blurry vision [1]	1	
Ishay (2021)	Pain [1]—Redness [1]—Blurry vision [1]	1	
Duran (2022)	Blurry vision [1]—Redness [1]—Headache [1]	1	
ElSheikh (2021)	Blurry vision [1]—Photophobia [1]	1	
Lee (2022)	Vision loss [1]	1	
Gedik (2022)	Pain [1]—Diminished vision [1]	1	
Goyal (2021)	Vision loss [1]	1	
Hébert (2022)	Vision loss [1]—Floaters [1]	1	
Hwang (2022)	Erythema [1]	1	
Jain (2021)	Pain [1]—Redness [1]	1	
K. Joo (2022)	Pain [1]—Blurry vision [1]—Headache [1]—Eyelid swelling [1]	1	
Kim (2022)	Vision loss [1]—Headache [1]	1	
Koong (2021)	Blurry vision [1]	1	
Papasavvas (2021)	Pain [1]	1	
Ding (2022)	Vision loss [1]	1	
Lee (2022)	Pain [1]—Blurry vision [1]	1	
Sai (2022)	Blurry vision [1]—Floaters [1]	1	
Matsuo (2022)	Blurry vision [1]	1	
Mishra (2021)	Pain [1]—Diminished vision [1]	1	
Mudie (2021)	Pain [1]—Photophobia [1]—Redness [1]—Vision loss [1]	1	
Pan (2021)	Vision loss [1]	1	
Papasavvas (2021)	Pain [1]—Diminished vision [1]—Photophobia [1]	1	
Reddy (2021)	Blurry vision [1]	1	
Renisi (2021)	Pain [1]—Redness [1]—Diminished vision [1]—Photophobia [1]	1	
Sangoram (2022)	Blurry vision [1]—Pain [1]	1	
Santiago (2021)	Redness [1]	1	
Saraceno (2021)	Vision loss [1]	1	
Singh (2022a)	Diminished vision [1]	1	
Yalçinkaya (2022)	Redness [1]	1	
Yamaguchi (2022)	Metamorphopsia [1]—Diminished vision [1]	1	
Shilo (2022)	Photophobia [1]—Vision loss [1]	1	
Kakarla (2022)	Blurry vision [1]—Headache [1]	1	
Numakura (2022)	Blurry vision [1]	1	
Murgova (2022)	Metamorphopsia [1]	1	
Patel (2022)	Pain [1]—Blurry vision [1]—Floaters [1]	1	
Lawson-Tovey (2022)	-	1	
Arora (2022)	Diminished vision [1]—Floaters [1]	2	
Choi (2022)	Diminished vision [3]	3	
Ortiz-Egea (2022)	Pain [2]—Redness [1]	2	
Nanji (2022)	Pain [2]—Redness [2]	2	
Pang (2022)	Blurry vision [1]	2	
Ren (2022)	Diminished vision [1]—Blurry vision [1]—Redness [1]—Pain [1]	2	
Cohen (2022)	Pain [1]—Photophobia [1]—Diminished vision [1]—Floaters [1]	4	
Aguiar (2022)	Redness [2]—Photophobia [2]—Pain [1]—Diminished vision [1]	2	
Ferreira (2022)	Vision loss [3]—Headache [4]—Blurry vision [1]—Hyperemia [1]	4	
Chen (2022b)	Blurry vision [5]—Redness [3]	5	
Chew (2022)	Blurry vision [6]—Redness [3]—Pain [1]	6	
Rallis (2022)	Diminished vision [7]—Pain [7]	7	
Li (2022)	-	9	
Sim (2022)	-	11	
Rabinovitch (2021)	Redness [21]—Pain [21]—Blurry vision [21]—Photophobia [21]—Photopsia [2]—Diminished vision [2]	21	
Bolletta (2021)	Blurry vision [12]—Redness [3]—Pain [2]—Photophobia [1]	13	
Ferrand (2022)	-	25	
Ozdede (2022)	-	5	
Testi (2022)	-	50	
Tomkins-Netzer (2022)	-	188	
Barda (2021)	-	26	
Singh (2022b)	Pain [270]—Redness [839]—Diminished vision [262]—Photophobia [95]—Floaters [21]—Lacrimation [22]	1094	
**Summary of Symptoms/Signs of VAU**			
**Presentation**	**Number**	**Total**	**%**
Central Scotoma	1	1211	0.08
Vision Loss	13	1211	1.07
Pain	53	1211	4.37
Erythema	2	1211	0.16
Photophobia	127	1211	10.48
Blurry vision	64	1211	5.28
Redness	884	1211	72.99
Diminished vision	285	1211	23.53
Headache	10	1211	0.82
Foreign Body Sensation	1	1211	0.08
Floaters	27	1211	2.22
Eyelid swelling	1	1211	0.08
Photopsia	2	1211	0.16
Metamorphopsia	2	1211	0.16

YOP: Year of Publication; VAU: Vaccine-Associated Uveitis; N: Number.

**Table 5 vaccines-11-00069-t005:** The type, laterality, course, location, onset, nature, and underlying cause of vaccine-associated uveitis.

Outcome	Category	Number	Total	%
**Type of VAU**				
	VKH	16	1476	1.08
	Choroiditis	9	1476	0.6
	Iridocyclitis	1	1476	0.06
	Iritis	2	1476	0.13
	Kerato-uveitis	1	1476	0.06
	Retinitis	2	1476	0.13
	Uveitis	1440	1476	97.56
	Retinochoroiditis	3	1476	0.2
	Pars planitis	2	1476	0.13
**Laterality**				
	Right	126	390	32.3
	Left	135	390	34.61
	Unilateral	303	390	77.69
	Bilateral	86	390	22.05
**Course**				
	Acute	234	326	71.77
	Chronic	92	326	28.22
**Location**				
	Anterior	799	1476	54.13
	Intermediate	14	1476	0.94
	Posterior	78	1476	5.28
	Panuveitis	148	1476	10.02
**Onset**				
	New-onset	244	349	69.92
	Reactivation	105	349	30.08
**Nature**				
	Autoimmune	4	299	1.34
	Granulomatous	6	299	2.01
	Inflammatory [non-infectious]	264	299	88.29
	Infectious	25	299	8.36
**Underlying Cause**				
	Behcet’s disease	4	245	1.63
	CMV	1	245	0.4
	HSV-1	9	245	3.67
	HZO	3	245	1.22
	JIA	4	245	1.63
	MIS-C	1	245	0.4
	Retinal vasculitis	1	245	0.4
	Sarcoidosis	3	245	1.22
	Toxoplasma	4	245	1.63
	VKH	18	245	7.34
	VZV	3	245	1.22
	Psoriasis	1	245	0.4
	Spondylarthritis	1	245	0.4
	Idiopathic	106	245	43.26
	HLA B27	12	245	4.89
	Fuchs heterochromic iridocyclitis	2	245	0.81
	Posner–Schlossman syndrome	1	245	0.4
**Duration from vaccination to uveitis attack (days)**				
	Mean—SD	9.61	8.07	
	Min—Max	1	42	
	Observations	108		

**Table 6 vaccines-11-00069-t006:** The outcomes and complications following the treatment of vaccine-associated uveitis.

Outcome	Category	Number	Total	%	Duration (days)
**Complications**					
	CME	2	83	2.41	60
	Choroidal depigmentation	2	83	2.41	14
	Inflammatory glaucoma	1	83	1.2	-
	Peripheral neovascularization	1	83	1.2	135
	Retinal necrosis	1	83	1.2	-
	Recurrence of choroidal thickening	1	83	1.2	21
	ME	2	83	2.41	180
	Uveitis exacerbation	1	83	1.2	-
	Vitritis	1	83	1.2	-
	Transient IOP elevation	3	83	3.61	-
	Nummular Corneal Lesions	3	83	3.61	-
**Treatment Outcome**					
	Complete Resolution	174	193	90.15	-
	Partial Improvement	19	193	9.85	-

CME: Cystoid Macular Edema; ME: Macular Edema; IOP: Intraocular Pressure.

## Data Availability

The data provided in this manuscript can be provided upon reasonable request by contacting the corresponding author.

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
