# Peer review of "The Characteristics of COVID-19 Vaccine-Associated Uveitis: A Summative Systematic Review"

_vaccines, 2022, doi:10.3390/vaccines11010069_

Round 1

Reviewer 1 Report

Please supplement the papers on the animal model of uveitis caused by covid-19 vaccination and its pathogenic mechanism.

Author Response

Thank you for your comment. Despite the availability of some studies that investigated the role of COVID-19 vaccination on ocular inflammation [Ng et al., 2022, Ophthalm Therp], to date, there are no evidence (particularly animal models) that investigate the role of COVID-19 vaccination on ocular inflammation in uveitis models/animals in particular.  

Reviewer 2 Report

This is an excellent systematic literature review on the occurrence of uveitis following COVID-19 vaccinations. The study is scientifically well-executed and thoroughly presented by the authors. 

Minor consideration: the authors reported "The majority of cases were documented in those who took the Pfizer BioNTech vaccine (77.9%), followed by Moderna (15.54%), and AstraZeneca (3.01%), respectively. "  and provided a possible explanation for that "This could be explained by the fact that Pfizer–BioNTech COVID-19 vaccine elicits an additional CD8 T-cell immune response, providing additional protection against SARS-CoV2 infection,  however, triggering autoimmune reactions". However, this finding can also be attributed to the fact that the vast majority of vaccines used globally were Pfizer-BioNtech. The authors should also acknowledge that and even better should try to adjust this result by the percentage of each type of vacine used globally. This will allow the readers to draw even better conclusions. 

Author Response

Thank you so much for your comment and helpful insights. We have added a new sentence to highlight this point as follows: "This could also be attributed to the dominance of Pfizer-BioNTech vaccine over other COVID-19 vaccine type in the number of administered doses. For instance, up to December 2022, 656.90 million Pfizer doses have been administered followed by Moderna (153.82 million), AstraZeneca (67.03 million), Jhonshon&Jhonson (18.93 million), Sinopharm (2.32 million), and Sputnik (1.85 million), respectively. Other vaccines (Sinovac, Novavax, and Covaxin) have been administered at a much lower rate (below 1 million doses)".

Thank you again for your tremendous help in improving the quality of our manuscript!

Reviewer 3 Report

I have one question to the Authors, what about COVID-19 in other organs?

Author Response

Thank you for your question. We truly appreciate your insight; however, our paper was focused on just a particular eye disorder (uveitis), given the heterogeneity in its causative agents (inflammatory, infectious, autoimmune, etc.). Additionally, the number of affected cases (VAU) has been increasing over time which grabbed our attention and highlighted the need to thoroughly discuss such observations in order to give ophthalmologists and ophthalmology trainees some guidance when they encounter such cases, so they would VAU as a part of their differential diagnosis for better management.

For these reasons, we just focused on this particular disease and did not go into other organs that have been affected by COVID-19 vaccines, which are numerous and should be discussed in future manuscripts (given the large sum of data in the literature in this regard).

Thank you again for your very interesting question!

Reviewer 4 Report

This manuscript reviewed and characterized the current status of COVID-19 Vaccine-Associated Uveitis (VAU) using different medical database. It is suitable for publication in this journal with minor revision. I only have several questions:

1.    It appears there is some gender difference (more female than male cases) in Covid-19 VAU, also Uveitis may have some relationship with genetic factors. I was wondering whether there is any relationship between Covid-19 VAU and genetic background, if possible, please give a brief discussion.

2.    The authors showed that the crude incidence rate of VAU was 0.57 cases per million doses of the COVID-19 Pfizer vaccine, if there is any data related to the incidence rate of other vaccinations than Covid-19 (such as HPV, Hepatitis B vaccine, etc) that cause VAU, please give some comparison.

3.    Is there any possibility there is increased incidence rate of Uveitis in the past 2-3 years due to Covid-19 vaccination?

Author Response

Comment 1: Thank you for your very insightful comment. Unfortunately, COVID-19 VAU has only been investigated and reported in numerous case reports and very few cohort studies. However, unfortunately, no animal models or other studies have investigated the link between VAU and genetic background or the pathogenicity of such an event. But given the increase in the number of reported cases (from being scarce case reports to being frequently encountered in clinical practices), research is currently being conducted to investigate this relationship, and our team is currently working on determining the pathogenicity of COVID-19 VAU in animal models.

Comment 2: Your questions and comments are truly insightful and will provide some ground for future research work. The crude incidence of COVID-19 VAU was reported based on the analysis of the VAERS registry; however, trying to determine the incidence rate of VAU following Hepatitis B vaccination (for example) would require separate analysis (assuming that such data are available) and a whole new manuscript. So, unfortunately, we will not be able to provide any sort of comparison for many reasons:

  • Our review is focused more on the clinical characteristics of COVID-19 VAU rather than the incidence over time or incidence as compared to other vaccines.
  • VAU has only been reported after Hep B vaccine in the past, in very few cases. And, analyzing a dataset as large as the VAERS registry needs more resources, time, analysis, and writing, which can be done in a separate manuscript.

Comment 3: That is another intriguing, yet very exciting question. This could possibly be true but this would require further research to confirm such a hypothesis. This can be done in a separate systematic review and meta-analysis of the prevalence of uveitis cases all over the world over the past 10 years, where the cases reported in each year are compared to other years to determine any trends in the prevalence of uveitis. However, even if the observed rate has increased during the COVID-19 pandemic, we cannot reach solid conclusions that the COVID-19 has caused such an increase due to the availability of a wide list of other possible causes for uveitis (i.e., trauma, infection, inflammation, etc.). Thank you again so much for your very insightful comments, and we will definitely put all of them into consideration in our future research! Your comments and help are much appreciated!

Round 2

Reviewer 1 Report

The revised manuscript has met the requirements of the journal.